# Inhibition of Ihh Reverses Temporomandibular Joint Osteoarthritis via a PTH1R Signaling Dependent Mechanism

**DOI:** 10.3390/ijms20153797

**Published:** 2019-08-03

**Authors:** Hongxu Yang, Mian Zhang, Qian Liu, Hongyun Zhang, Jing Zhang, Lei Lu, Mianjiao Xie, Di Chen, Meiqing Wang

**Affiliations:** 1State Key Laboratory of Military Stomatology & National Clinical Research Center for Oral Diseases & Shaanxi International Joint Research Center for Oral Diseases, Department of Oral Anatomy and Physiology and TMD, School of Stomatology, the Fourth Military Medical University, Xi’an 710032, China; 2Department of Biochemistry, Rush University Medical Center, Chicago, IL 60612, USA

**Keywords:** chondrocyte, Indian hedgehog, osteoarthritis, parathyroid hormone related protein receptor 1, temporomandibular joint, terminal differentiation

## Abstract

The temporomandibular joint (TMJ), which is biomechanically related to dental occlusion, is often insulted by osteoarthritis (OA). This study was conducted to clarify the relationship between Indian hedgehog (Ihh) and parathyroid hormone receptor 1 (PTH1R) signaling in modulating the enhanced chondrocyte terminal differentiation in dental stimulated TMJ osteoarthritic cartilage. A gain- and loss-of-function strategy was used in an in vitro model in which fluid flow shear stress (FFSS) was applied, and in an in vivo model in which the unilateral anterior cross-bite (UAC) stimulation was adopted. Ihh and PTH1R signaling was modulated through treating the isolated chondrocytes with inhibitor/activator and via deleting *Smoothened* (*Smo*) and/or *Pth1r* genes in mice with the promoter gene of type 2 collagen (*Col2-CreER*) in the tamoxifen-inducible pattern. We found that both FFSS and UAC stimulation promoted the deep zone chondrocytes to undergo terminal differentiation, while cells in the superficial zone were robust. We demonstrated that the terminal differentiation process in deep zone chondrocytes promoted by FFSS and UAC was mediated by the enhanced Ihh signaling and declined PTH1R expression. The FFSS-promoted terminal differentiation was suppressed by administration of the Ihh inhibitor or PTH1R activator. The UAC-promoted chondrocytes terminal differentiation and OA-like lesions were rescued in *Smo* knockout, but were enhanced in *Pth1r* knockout mice. Importantly, the relieving effect of *Smo* knockout mice was attenuated when *Pth1r* knockout was also applied. Our data suggest a chondrocyte protective effect of suppressing Ihh signaling in TMJ OA cartilage which is dependent on PTH1R signaling.

## 1. Introduction

Osteoarthritis (OA) is a type of joint problem, which is often caused by biomechanical factors, and cartilage is the most frequently involved site [1]. Cells located in articular cartilage are differentiated from early to late stage [2]. The superficial zone cells are proliferative, and the deep zone chondrocytes are hypertrophic, from which stage the cells undergo terminal differentiation [3]. One of the main characteristics of OA is accelerated cellular differentiation [4]. Generally, in OA cartilage, chondrocytes recapitulate a course that resembles the process of chondrocyte terminal differentiation in terms of phenotypic and gene expression changes in the growth plate [5]. Cellular differentiation in the growth plate has been extensively reported [6]. However, little is known about the molecular mechanisms of the biomechanically accelerated chondrocyte differentiation in osteoarthritic degenerative cartilage.

Many signaling pathways are involved in modulating chondrocyte differentiation and hypertrophy in the growth plate, among which the Indian hedgehog (Ihh) and parathyroid hormone-related peptide (PTHrP) feedback loops play a key role [7]. Ihh, a member of the hedgehog family of proteins, releases its inhibition on smoothened (Smo) and promotes chondrocytic to hypertrophic terminal differentiation through binding to the patched-1 receptor [8]. PTHrP, in contrast, prevents chondrocytes from premature differentiation by activating parathyroid hormone/PTHrP receptor 1 (PTH1R), the only receptor of PTHrP ligand [9,10]. Ihh facilitates OA development because the upregulation of Ihh promotes the expression of genes known to be involved in cartilage degeneration, while the inhibition of Ihh attenuates OA severity [3,11]. Such an effect of Ihh is achieved most likely through PTHrP signaling because the overexpression of PTHrP could reverse mechanical strain-induced chondrocyte hypertrophy [12,13,14,15]. However, the relationship between Ihh and PTH1R in the enhanced terminal cellular differentiation process during OA progression remains unclear.

Chondrocytes express specific arrays of gene products demarcating the various stages of chondrocyte development when they differentiate from proliferative into hypertrophic cells [16]. At the early differentiation stage, chondrocytes express sex determining region Y-box 9 (Sox9), followed by type II collagen (Col2a1) and proteoglycan (PG) [17]. When chondrocytes reach the hypertrophic stage and subsequently endochondral ossification, type X collagen (Col-X), alkaline phosphatase (ALP), and matrix metalloprotease-13 (MMP-13) are expressed [6]. In OA cartilage, the expression levels of Col-X, MMP-13 and ALP are often increased [5,7]. By testing the markers of different differentiation stages, together with histomorphology, the role of Ihh and PTH1R signaling in experimental degenerative articular cartilage could be evaluated through adopting the genetic mouse models [18,19,20].

Temporomandibular joint (TMJ), a site that is frequently involved in OA, is biomechanically related to dental occlusion [21]. Studies have indicated that TMJ OA induced by interrupted dental occlusion undergoes different pathological processes as compared to knee OA [22]. Recently, we created a rodent model by exposing the animals to an aberrant biomechanical dental occlusion, termed unilateral anterior crossbite (UAC), which simulated clinical abnormal dental occlusion and caused a series of TMJ OA-like changes without surgical damage [23,24,25,26,27,28,29,30]. Our latest research suggested that one of the typical changes in chondrocytes in UAC rat TMJ OA cartilage was abnormal differentiation, starting from the injured deep zone and then expanding to the superficial zone [26]. We also established an in vitro model through which the responses of the isolated chondrocytes to the flow fluid shear stress (FFSS) were investigated [26,31,32]. Using both the in vivo UAC model and in vitro FFSS model, as well as by exposing the transgenic mice to UAC, the role of Ihh and PTH1R signaling in the enhanced process of TMJ chondrocyte terminal differentiation was explored. The aim of this study was to determine whether modulating both signals could alter dental biomechanically enhanced chondrocyte terminal differentiation in TMJ cartilage and to elucidate the relationship of these two signaling pathways during this process.

## 2. Results

### 2.1. UAC Stimulated Chondrocyte Terminal Differentiation with the Modulation of Ihh and PTH1R Signaling

The rat TMJ condylar cartilage typically contains four layers, i.e., the fibrous, proliferative, prehypertrophic, and hypertrophic layer [33] (Figure 1a). Consistent with our previous results, OA-like lesions in TMJ cartilage were induced by UAC, displayed as a decreased thickness of the deep zone, and included predominantly prehypertrophic and hypertrophic layers, and a reduction of the cartilage matrix from two to eight weeks (Figure 1b,c). The thickness of the superficial zone, which was mainly composed of the fibrous and proliferative layers, was the least insulted, and the condylar surface was intact (Figure 1b,c). The expression levels of chondrogenic genes, including *Acan* and *Col2a1*, were downregulated in UAC group (Figure 1d). However, the levels of the markers related to chondrocyte terminal differentiation, such as *Opn*, *Col10a1*, *Alp*, *Mmp13*, and *Runx2*, were upregulated (Figure 1e). Further, Western blot revealed the aggravated differentiation by upregulation of ALP and Col-X (Figure 1f).

Impressively, Ihh positive cells in both UAC and sham control groups were located in the pre-hypertrophic and hypertrophic layers, and the percentage of Ihh positive cells in the UAC group increased from two to eight weeks as revealed by immunohistochemistry (IHC) (Figure 2a). In contrast, PTHrP was expressed predominantly in the proliferative layers in both UAC and sham control groups, and the percentage of PTHrP-positive cells was increased from four to eight weeks in the UAC group (Figure 2b). Upregulation of the expression of Ihh and PTHrP in the UAC group were confirmed by WB and qPCR assays (Figure 2d,e). Interestingly, both Smo and PTH1R were located in the prehypertrophic and hypertrophic layers (Figure 2c and Appendix A). The expression levels of Smo was not changed in the UAC group, while the expression of PTH1R was downregulated in the UAC group from two to eight weeks, revealed by IHC and WB (Figure 2d,e). These results indicate that UAC stimulation leads to terminal differentiation of deep zone chondrocytes, and such an effect may be promoted by the Ihh signaling via reducing the number of PTH1R positive cells.

### 2.2. FFSS Stimulated Deep Zone Chondrocytes Terminal Differentiation with Activated Ihh Signaling but Suppressed PTH1R Expression

Cells from superficial and deep zones of condylar cartilage were separately isolated from three-week-old Sprague-Dawley (SD) rats under the microscope (Appendix A). The cultured cells were exposed to 4 and 24 dyn/cm^2^ FFSS for 1 h for biomechanical stimulation in vitro. Fluorescence staining with dead/live dye revealed that both 4 and 24 dyn/cm^2^ FFSS induced deep zone cells’ death but only 24 dyn/cm^2^ FFSS induced superficial zone cells’ death (Appendix A). In superficial zone cells, FFSS at 24 dyn/cm^2^, but not at 4 dyn/cm^2^, downregulated the mRNA expression of *Acan*, *Col2a1*, and *Prg4* (Appendix A) and the protein expression of Sox9, while neither of these two levels of FFSS had significant effects on the protein expression of Col-X, which was hardly expressed in this zone (Appendix A). In deep zone cells, FFSS at both 4 and 24 dyn/cm^2^ downregulated the mRNA expression of *Acan* and *Col2a1* and upregulated the protein expression of Col-X, and did not affect the mRNA expression of *Prg4* or the protein expression of Sox9, both of which were minimally expressed herein (Appendix A). The mRNA and/or protein expression of the differentiation related markers were all upregulated by FFSS stimulation in deep zone cells, while they were minimally expressed and remained unchanged in the superficial-zone cells (Appendix A).

The immunofluorescence (IF) data showed that the superficial zone cells expressed high levels of PTHrP but low levels of Ihh. In contrast, the deep zone cells were mostly PTHrP-negative but Ihh-positive (Figure 3a–c). Both Smo and PTH1R were highly expressed in the deep zone but not in the superficial zone. The expression level of PTHrP was strengthened by the 24 dyn/cm^2^ FFSS treatment. The expression level of Ihh was upregulated while that of PTH1R was downregulated by 4 and 24 dyn/cm^2^ FFSS. The expression level of Smo was not affected by 4 and 24 dyn/cm^2^ FFSS (Figure 3d–f).

Taken together, our data indicate that UAC in vivo and FFSS in vitro promoted the terminal differentiation of deep zone chondrocytes with enhanced Ihh signaling but reduced PTH1R expression, although it upregulated the production of PTHrP in superficial zone cells.

### 2.3. Suppressing Ihh Signaling Rescued FFSS and UAC Stimulated Chondrocytes Terminal Differentiation

To gain insights into the effect of suppressing Ihh signaling to the FFSS treated deep zone chondrocyte differentiation, we added vismodegib or purmorphamine, the antagonist or activator of Smo and the ligand of Ihh, respectively, to the culture medium before the application of 24 dyn/cm^2^ FFSS. Vismodegib did not change the Ihh expression but prevented the downregulation of PTH1R expression induced by FFSS (Appendix A). Vismodegib reversed the FFSS-stimulated increased expression of the markers related to chondrocyte terminal differentiation, such as *Opn*, *Col10a1*, *Alp*, *Mmp13*, and *Runx2* (Appendix A) and the FFSS-suppressed expression of cartilage matrix genes, including *Acan* and *Col2a1* (Appendix A). In contrast, purmorphamine reduced the expression level of PTH1R (Appendix A) and promoted the expression of the terminal differentiation related markers and cartilage matrix. Similar to vismodegib, purmorphamine did not change the expression of Ihh (Appendix A).

We next generated cartilage-specific and tamoxifen-inducible *Smo* gene knockout mice and exposed them to UAC (Appendix A). At six weeks of age, *Col2-CreER;Smo*^fl/fl^ mice were treated with tamoxifen (TM) to delete the *Smo* gene sustained at three and seven weeks [34]. *Smo*^fl/fl^ littermates were used as a blank control and *Smo*^fl/fl^ mice with TM and *Col2-CreER;Smo*^fl/fl^ mice were used as negative controls. Our data showed that Gli1, which are the readout of the Ihh signaling pathway, were transcriptionally inhibited in *Smo* knockout mice (*Smo*-KO). In the *Smo*-KO mice, the deep zone thickness was increased compared with three control groups, and the UAC induced OA-like lesions were significantly suppressed compared with the UAC treated mice in the three control groups. In the three control groups the UAC induced OA-like lesions were identical (Appendix A). We then used the data from *Smo*^fl/fl^ group as an example of the control in the following descriptions.

In the *Smo*^fl/fl^ control group, like what were observed in the wild type rats described above, UAC induced OA-like lesions were obvious, showing as enhanced chondrocyte terminal differentiation of the deep zone cells and reduced cartilage matrix and aggrecan (Figure 4a,b and Appendix A). The differentiation-related markers including MMP13, Col-X, and ALP were increased, accompanied with the upregulation of Ihh and downregulation of PTH1R (Figure 4c–g and Appendix A). In the *Smo*-KO group, however, the UAC induced cartilage thickness reduction and matrix degradation, and the UAC stimulated changes in the expression of aggrecan, MMP13, Col-X, and ALP were all significantly suppressed (Figure 4a–e and Appendix A), although the expression level of Ihh was not altered (Figure 4f and Appendix A). Importantly, the expression of PTH1R was increased in the *Smo*-KO group versus the *Smo*^fl/fl^ control group (Figure 4g and Appendix A).

Finally, we injected vismodegib or purmorphamine, respectively, into the TMJ local area to demonstrate the effect of abrogating Ihh signaling on UAC-enhanced terminal differentiation of deep zone chondrocytes. Vismodegib (1 μM) or purmorphamine (1 μM) was injected to UAC treated rats every other day from four to eight weeks after UAC, for a total of 14 times within 4 weeks. No significant inflammatory responses were noticed in the injection samples. The data showed that vismodegib inhibited the Ihh signaling and rescued the UAC-induced cartilage lesions (Figure 5a–c). Meanwhile, there was an upregulation of PTH1R (Figure 5d,e). In contrast, purmorphamine elicited an opposite effect because it caused an aggravation effect of OA lesions (Figure 5a–c). The expression levels of PTH1R were downregulated (Figure 5d,e). The vismodegib, purmorphamine, and vehicle showed no influences on the Ihh expression (Figure 5d–f).

These data imply that the suppressing effect on chondrocyte terminal differentiation via inhibiting Ihh signaling under FFSS or UAC was associated with upregulation of PTH1R.

### 2.4. Reduction of PTH1R directly Promoted FFSS and UAC Stimulated Chondrocyte Terminal Differentiation

To detect whether the reduced PTH1R expression played a key role in FFSS-promoted terminal differentiation, we downregulated the expression of *Pth1r* in the deep zone cells by siRNA. The mRNA expression of *Pth1r* and its related downstream readout of proteinkinase A (PKA) were successfully downregulated by the siRNA (Figure 6a,b). As expected, downregulation of PTH1R reduced the expression of *Acan* and *Col2a1* and increased the expression of chondrocyte differentiation markers (Figure 6c–e). Interestingly, downregulation of PTH1R increased the FFSS-stimulated expression of Ihh (Figure 6e) and the markers related to chondrocytes terminal differentiation, but aggravated the reduction of chondrogenic genes (Figure 6a–e).

We then pretreated the deep zone cells with 1 μM PTHrP7.34 or PTHrP1.34, the invalid or effective ligand of PTH1R, respectively, before exposing them to the 24 dyn/cm^2^ FFSS. Data indicated that pretreatment with PTHrP7.34 had no influence on the cells, with or without FFSS stimulation (Appendix A), while pretreatment with PTHrP1.34 rescued the FFSS-induced downregulation of the *Acan* and *Col2a1* expression and reversed the terminal differentiation at the protein and/or mRNA levels. Pretreatment with PTHrP1.34 showed no effect on the cells without FFSS treatment (Appendix A). Neither PTHrP7.34 nor PTHrP1.34 showed any effects on the FFSS-downregulated PTH1R expression or PKA expression (Appendix A). Interestingly, PTHrP1.34 showed a suppressing effect on the FFSS-stimulated Ihh expression (Appendix A).

We next generated cartilage-specific and tamoxifen-inducible *Pth1r* gene knockout mice and exposed them to UAC (Appendix A). At six weeks of age, *Col2-CreER;Pth1r*^fl/fl^ mice were treated with TM to delete the *Pth1r* gene as it was reported [35]. *Pth1r*^fl/fl^ littermates were used as a blank control, and *Pth1r*^fl/fl^ with TM and *Col2-CreER;Pth1r*^fl/fl^ mice were used as negative controls (Appendix A). IHC staining revealed that PTH1R expression was transcriptionally inhibited in *Pth1r* knockout mice (*Pth1r*-KO), while no changes were detected in the three control groups under the *flox* transgene and tamoxifen treatment. There were no histomorphological differences between the three genetic control groups (Appendix A). Knockout of *Pth1r* could decrease the deep zone thickness of cartilage as shown in the sham group (Appendix A). Moreover, UAC decreased the expression of PTH1R and aggravated the loss of the deep zone thickness compared with the sham group (Appendix A). In the following descriptions, we used the data from the *Pth1r*^fl/fl^ group as an example of the control.

In the *Pth1r*^fl/fl^ control group, UAC promoted the deep zone chondrocyte terminal differentiation, displaying decreased thickness of the deep zone cartilage, and reduced matrix amount and aggrecan expression (Figure 7a,b and Appendix A). The expression of Col-X, MMP13, and ALP were almost unobservable in sham group, while UAC stimulated their expression and increased the percentages of the positive cells which were located at the deep zone cartilage (Figure 7c–e and Appendix A). The percentage of the Ihh positive cells which were located in the deep zone layer was also increased by UAC in the *Pth1r*^fl/fl^ control group (Figure 7f and Appendix A). Notably, knockout of *Pth1r* showed a similar phenotype (loss of cartilage matrix) as UAC treated *Pth1r*^fl/fl^ control mice (Figure 7a,b and Appendix A) and promoted chondrocyte terminal differentiation (Figure 7c–e and Appendix A). Interestingly, deletion of *Pth1r* promoted the expression level of Ihh in deep zone cartilage (Figure 7f and Appendix A). When UAC was delivered to the *Pth1r* knockout mice, the terminal differentiation was enhanced compared with the UAC treated *Pthr1*^fl/fl^ control mice and the *Pth1r* knockout mice without UAC treatment (Figure 7 and Appendix A). In all groups, the superficial zone remained intact.

These results demonstrate that a decline in PTH1R expression promoted FFSS- and UAC-induced terminal differentiation of the chondrocytes in TMJ cartilage. The relation of PTH1R expression and Ihh signaling in this process still remained obscure because they were accompanied with an increased Ihh expression when *Pth1r* was knockout. We then converted the *Smo* and *Pth1r* double knockout experiments.

### 2.5. Inhibition of Ihh Prevented UAC Induced Chondrocyte Terminal Differentiation through a PTH1R Dependent Mechanism

To clarify the relationship of the two signaling molecules in the process, we generated *Col2-CreER;Smo*^fl/fl^-*Pth1r*^fl/fl^ mice to gain *Pth1r* and *Smo* double knockout mice (*Pth1r-Smo*-double KO). The *Smo*^fl/fl^-*Pth1r*^fl/fl^ littermates and the *Smo* knockout mice were used as controls. There were compared with the *Smo* knockout mice at seven weeks. The *Smo*^fl/fl^-*Pth1r*^fl/fl^ littermates and the *Smo* knockout mice were used as controls. In the sham group, *Smo* knockout inhibits Ihh signaling as previous results indicated, leading to the increase of the cartilage thickness and aggrecan expression compared with the *Smo*^fl/fl^-*Pth1r*^fl/fl^ control group (Figure 8a–d first and second column). The data showed that double knockout mice had a phenotype that was very similar to that of *Pth1r* knockout mice, i.e., the rescue effect of *Smo* deletion on cartilage thickness, matrix, and aggrecan expression was suppressed in the sham group (Figure 8a–d second and third column). Meanwhile, the double knockout of *Smo* and *Pth1r* promoted cartilage terminal differentiation by the upregulation of Col-X and ALP (Figure 8e–h second and third column). It was also similar to *Pth1r* knockout in that the expression of Ihh was increased (Figure 8i,j second and third column). Such an effect became more obvious when mice were exposed to UAC stimulation. The rescue effect of *Smo* deletion on UAC-induced TMJ OA-like lesions was abrogated in *Pth1r* and *Smo* double knockout mice, as evaluated by the cell size, cartilage thickness, cartilage matrix, and expression levels of aggrecan (Figure 8a–d fifth and sixth column). Further, double knockout could increase the expression of terminal differentiation markers such as Col-X and ALP accompanied with the upregulation of Ihh (Figure 8e–j fifth and sixth column). These findings suggest that the preventive effect of Ihh signaling suppression on the terminal differentiation of chondrocytes induced mechanically or biomechanically is dependent on PTH1R signaling.

## 3. Discussion

Accumulating evidence demonstrates that the structural and cellular components of mandibular condylar cartilage have the capacity to adapt to the joint function link under dental occlusion loading [36,37]. The present data confirm our previous findings that deep zone chondrocytes respond to UAC, as well as FFSS, by terminal differentiation in addition to death [26]. The data also indicate that both Ihh and PTH1R signaling were involved in this process. Ihh signaling plays a role in enhancing deep zone chondrocyte terminal differentiation, while PTH1R signaling shows an opposite effect. Thus, modulating this signaling could suppress the terminal differentiation of the TMJ chondrocytes and attenuate the dental stimulated TMJ OA progression.

The promotion effect of Ihh signaling on chondrocytes differentiation has been extensively reported in the growth plate [38]. In the present disease model, inhibition of Ihh signaling rescued UAC-stimulated chondrocyte terminal differentiation and cartilage degradation, while the inhibition of PTH1R expression enhanced this effect. Importantly, the deletion of the *Smo* gene showed a promotion effect on the number of PTH1R-positive cells, implying that the rescue effect of inhibiting Ihh signaling on OA lesions was due to the preservation of PTH1R positive cells. This assumption was confirmed in double knockout mice. Additional deletion of *Pth1r* abrogated the rescue effect of *Smo* deletion on UAC-stimulated cartilage degradation. Obviously, the rescue effect of Ihh inhibition on OA lesions is PTH1R dependent.

Both Ihh and PTHrP are activated when chondrocytes are biomechanically stimulated [39]. In the growth plate, there was a suppression effect of the enhanced PTHrP signaling on the function of Ihh signaling. Such an effect is generally described as a negative feedback loop that mediates the proliferation and hypertrophy of the chondrocytes [40,41]. PTH1R is the only known receptor of PTHrP [10]. Emerging evidence reveals that reducing the expression of PTH1R in articular cartilage accompanied with the development of OA exists, but the effect of it has not been demonstrated [10,42]. We presently disclosed that sufficient PTH1R positive cells in *Smo* knockout mice were pivotal for decelerating chondrocyte differentiation in UAC stimulated OA cartilage. Via increasing of the PTH1R positive cells, deletion of Smo successfully enhanced PTHrP signaling.

Chondrocytes in cartilage display depth-dependent phenotypes that endow the cartilage with zonal biomechanical properties [43]. Emerging evidence highlights that the fibrous layer lining the condyle surface is able to resist shear force, while deep zone cells are less resistant to the induction of cell death [44,45,46]. Deep zone chondrocytes respond to the shear force by terminal differentiation, showing increased expression levels of Col-X, ALP, and MMP13 accompanied by decreased cartilage thickness. Technically, we are not able to further separate chondrocytes in different zones, such as between prehypertrophic and hypertrophic layers [47]. We used the laser capture technique, which could only distinguish superficial zone cells that were demonstrated to express *Sox9* and *Prg4* and deep zone cells that were positive for Ihh and Col-X. Thus, based on the present data, we could not confirm whether Ihh, Smo, and PTH1R were expressed in the same cells. However, they were expressed in the same subpopulation of cells, i.e., cells located in the zone beneath the superficial layer. The same data were revealed by IHC staining, localizing Ihh, Smo, and PTH1R in the deep zone chondrocytes. Thus, by targeting the same group of cells, Ihh signaling promotes chondrocyte differentiation while PTH1R signaling prevents such a process, as previously reported [48].

The cellular zonal boundary of mouse TMJ cartilage is not as morphologically typical as that in rats. However, our previous study revealed that the TMJs of rats and mice respond to UAC stimulation in a very similar manner considering the cartilage and subchondral bone phenotypes and chondrocyte terminal differentiation [23,24,25,26,27,28,29,30,31,32]. Briefly, the UAC-induced TMJ OA-like changes predominantly affected the deep zone cartilage. The surface of the cartilage in both rats and mice was intact when they were exposed to UAC. We used rats to extract sufficient primary chondrocytes from the different layers and mice to generate the conditional cartilage-specific knockout litters with the promoter gene of the type 2 collagen (*Col2-CreER*), the marker of mature chondrocytes among deep-zone cells [49].

Articular cavity injection of drugs or inhibitors are adopted in clinical treatment for OA disease [9,23,25]. Here, we injected the Ihh inhibitor into the local area of rat TMJ to cure UAC-induced OA. Our results were consistent with the trend of in vitro FFSS stimulation in that the inhibition of Ihh could effectively inhibit the deep zone chondrocytes terminal differentiation. We used the insulin syringe to execute the local injection. No observable inflammatory reactions were noticed and all of our samples received the continuous injection 14 times, as we previous reported [23].

Taken together, our findings demonstrated both PTHrP and Ihh signaling have modulatory effects on chondrocyte terminal differentiation (Figure 9). Inhibiting one of them will impact on the other. However, the rescue effect of inhibiting Ihh signaling in regulation of the dental biomechanically stimulated TMJ chondrocytes’ terminal differentiation is PTH1R expression dependent. Our results provide novel insights into new therapeutic strategies for TMJ OA by protecting chondrocytes through promotion of the feedback loop between PTH1R and Ihh signaling. Enhancement of such an effect could be the primary requirement of OA treatment.

## 4. Materials and Methods

### 4.1. Experimental Animals

Animal care and all animal procedures were performed according to the institutional ARRIVE guidelines and were approved by the Ethics Committee of the Fourth Military Medical University (SYXK2015-001). All surgeries were performed under sodium pentobarbital anesthesia, and all efforts were made to minimize suffering. Thirty-six 6-week-old female Sprague-Dawley (SD) rats (weight 140–160 g) were provided by the animal center of the Fourth Military Medical University in Xi’an, China. The animals had free access to food and water and were acclimatized to the laboratory conditions (12 h light/dark cycle; 22 ± 1 °C room temperature) 1 week before experimentation. The rats were equally divided into sham and UAC groups at 2, 4, and 8 week time-points (*n* = 6).

To generate tamoxifen (TM)-inducible and cartilage-specific *Smo* knockout mice (*Col2-CreER;Smo*^fl/fl^), we crossed *Col2-CreER* mice with *Smo*^fl/fl^ mice [34]. At 6 weeks of age, *Col2-CreER;Smo*^fl/fl^ mice were treated with TM for the deletion of the *Smo* gene. *Smo*^fl/fl^ littermates were used as blank controls, and *Smo*^fl/fl^ with TM and *Col2-CreER;Smo*^fl/fl^ mice were used as negative controls. These four groups received the UAC or sham operation for 3 and 7 weeks. To generate tamoxifen (TM)-inducible and cartilage-specific *Pth1r* knockout mice (*Col2-CreER;Pth1r*^fl/fl^), we crossed *Col2-CreER* mice with *Pth1r*^l/fl^ mice [35]. The groups were the same as the *Smo* knockout mice that received the UAC or sham operation for 3 and 7 weeks. Tamoxifen (TM)-inducible and cartilage-specific *Smo* and *Pth1r* double knockout mice (*Col2-CreER;Smo*^fl/fl^-*Pth1r*^fl/fl^) were also generated, and *Col2-CreER;Smo*^fl/fl^-*Pth1r*^fl/fl^ mice were used as controls.

### 4.2. Tamoxifen Administration and Genotyping

Intraperitoneal injection of tamoxifen (T5648, Sigma-Aldrich, Darmstadt, Germany, 0.1 mg/g of body weight) was performed daily for 5 consecutive days from 1 day before the UAC operation and routine genotyping of mouse tail DNA was confirmed by RT-PCR. The primers used are shown in Appendix A for genotyping.

### 4.3. UAC Model and Tissue Preparation

The UAC operation was applied to the UAC group and the injection group rats, as previously described by our group [24,31,32]. Briefly, a section of a metal tube was adhered to the left maxillary incisor, and a curved section of a metal tube was adhered to the left mandibular incisor under deep anesthesia with intraperitoneal 1% sodium pentobarbital. The mandibular tube was curved to form a 135° labially inclined occlusal plate to create a crossbite relationship with the maxillary-tubed incisor. The technique of the TMJ local area injection was described in our previous report [23]. After UAC stimulation, the rats were laid sidelong after induction of deep anesthesia. Pentobarbital overdose was provided to euthanize all rats, and the TMJs were sampled (*n* = 6). Similar to our previous studies [24,28,31], there were no significant differences between the left- and right-side samples in the histomorphology or molecular properties. Six left TMJs from each group were sampled based on histochemical, Safranin O, and immunohistochemical staining (*n* = 6). Six right TMJs from each group were divided into three separate samples and used for analyses of mRNA by qRT-PCR assays and protein levels by Western blotting (*n* = 3).

### 4.4. Primary Chondrocyte Culture and Fluid Flow Shear Stress

Superficial zone and deep zone primary chondrocytes were isolated from the 3-week-old rat mandibular condylar cartilage through enzymatic digestion, for 20 min with 0.25% parenzyme (Hyclone), followed by 2.5 h with 0.2% type II collagenase (Gibco) in DMEM (Hyclone). The isolated cells from two zones were collected by brief centrifugation and were then resuspended in DMEM supplemented with 10% (*v*/*v*) FBS, 50 mg/mL streptomycin, and 50 unit/mL penicillin (Hyclone). The superficial and deep zone chondrocytes were plated in culture dishes at a density of 5 × 10^5^ cells/cm^2^. The medium was replaced every 2 days, and the cells reached confluence and adherence in culture for further experiments.

Fluid flow shear stress (FFSS) treatment was applied to the cultured cells using a Flexcell Streamer System (FX4000, Flexcell International, Co., Burlington, VT, USA). Briefly, the cells were cultured on the type I collagen-coated glass slides (Flexcell) for 24 h before exposure to an intensity of 4 or 24 dyn/cm^2^ for 1 h. Pretreatment with vismodegib, purmorphamine, PTHrP7.34, or PTHrP1.34 was applied to the culture medium 2 h before the application of FFSS.

### 4.5. siRNA Transfection for Pth1r Knockdown

The *Pth1r* siRNA was designed by Biomics (Nantong, China) to downregulate the expression. The sequences are siRNA-1353: Forward GGCAGAUCCAGAUGCAUUATT; reverse UAAUGCAUCUGGAUCUGCCTT. siRNA-435: Forward GUCCCGAUUACAUUUAUGATT; reverse UCAUAAAUGUAAUCGGGACTT. siRNA-917: Forward CUGGCUACCAACUACUACUTT; reverse AGUAGUAGUUGGUAGCCAGTT. Transfection of the deep zone cells with siRNA was performed according to the manufacturer′s protocol. The siRNA/lipofectamine mixture was transferred into 6-well plates for 8 h at 37 °C. Following replacement of the culture medium, the cells were incubated in α-MEM. The effect of knockdown was verified by qPCR and Western blot.

### 4.6. Histomorphology and Immunohistochemical Staining

TMJ blocks were fixed, decalcified, dehydrated, and embedded using conventional methods. The 4 µm thick sections were used for histomorphology and immunohistochemical staining. The serial sections were stained with Safranin O as previously reported in order to observe histological and proteoglycan changes in the articular cartilage [26,32]. IHC staining with anti-Ihh antibody (13388-1-AP, Proteintech, Philadelphia, IL, USA), anti-Smo antibody (ab113438, Abcam, Cambridge, UK), anti-PTHrP antibody (sc-20728, Santa Cruz, CA, USA), anti-PTH1R antibody (sc-20749, Santa Cruz, CA, USA), anti-Aggrecan antibody (ab36861, Abcam, Cambridge, UK), anti-ALP antibody (ab108337, Abcam, Cambridge, UK), anti-Collagen X antibody (ab58632, Abcam), anti-MMP13 antibody (18165-1-AP, Proteintech), and anti-Gli1 antibody (ab151796, Abcam) was performed as a standard, three-step, avidin-biotin complex staining procedure. Images were captured by a Leica light microscope (Leica 2500, Hessen, Germany). For negative controls, nonimmune goat serum was substituted for the primary antibody. The Sox9 expression in primary chondrocytes was detected with anti-Sox9 antibody Alexa Fluor @647 (ab196184, Abcam).

### 4.7. Immunoblotting, PCR and qRT-PCR

The protein and total RNA were extracted from condylar cartilage and cultured cells with Tripure (Roche) according to the manufacturer’s instructions. The protein was lysed with a protease inhibitor cocktail and 1% sodium dodecyl sulfate (SDS) and the protein content of the lysates was determined using Bicinchoninic Acid Kit for protein determination (Pierce Rockford, Waltham, MA USA) with BSA as the standard. The protein samples (30 μg) were resolved on 10% SDS-PAGE gels, transferred onto nitrocellulose membranes (Bio-Rad, Hercules, CA, USA) at 4 °C and 300 mA for 1 h and then blocked with 5% nonfat milk for 1 h. The membranes were incubated with primary antibodies overnight for anti-Ihh antibody (13388-1-AP, Proteintech), anti-Smo antibody (ab113438, Abcam, Cambridge, UK), anti-PTHrP antibody (sc-20728, Santa Cruz, CA, USA), anti-PTH1R antibody (sc-20749, Santa Cruz, CA, USA), anti-ALP antibody (ab108337, Abcam, Cambridge, UK), and anti-Collagen X antibody (ab58632, Abcam, Cambridge, UK). After rinsing, the membrane was incubated with horseradish peroxidase (HRP)-conjugated IgG secondary antibodies (Zhongshan Golden Bridge, Xi’an, China). Protein bands were detected using an enhanced chemiluminescence (ECL) system (Bio-Rad).

cDNA was generated via reverse-transcriptase using a PrimeScript™ RT Reagent Kit Perfect Real Time (RR036A, TaKaRa Biotechnology, Tokyo, Japan) with 1 μg of total RNA in 20 μL volumes at 37 °C for 20 min. SYBR^@^
*Premix Ex Tag*™ II (RR820A, TaKaRa) was used for quantitative real-time PCR (qPCR) analysis to evaluate gene expression, which was analyzed with the Bio-Rad real-time PCR system (Bio-Rad) according to the manufacturer’s recommendations, with *Gapdh* serving as the internal control. The amount of target mRNA was calculated using the formula 2^−ΔΔCT^. The primers used for rats are shown in Appendix A.

### 4.8. Statistical Analysis

Statistical analysis was accomplished using Graphpad Prism 7 software (GraphPad company, San Diego, CA, USA). Data are expressed as the means ± standard deviation for each group. The normality of the data distribution was tested using the Shapiro–Wilk test with 95% confidence and Levene’s test was used to assess homogeneity of variance. The percentages of degraded cartilage areas were compared by the nonparametric Kruskale–Wallis test and Mann–Whitney U test with the SPSS 22.0 software (SPSS Inc., Chicago, IL, USA). For all quantification at 2, 4, and 8 weeks, significant differences were identified by two-way analysis of variance. The Sidak test was performed for correct multiple comparisons using statistical hypothesis testing between every time point. For multiple comparisons of three or more groups, one-way analysis of variance with Tukey’s post hoc test was used, and Dunnett’s *t* method was used to evaluate the statistical significance if the homogeneity of variance was not equal. Statistical power was calculated by SAS 9.3 software (SAS Institute, Cary, NC, USA). *p* values less than 0.05 were considered statistically significant for all tests.

## Figures and Tables

**Figure 1 ijms-20-03797-f001:**
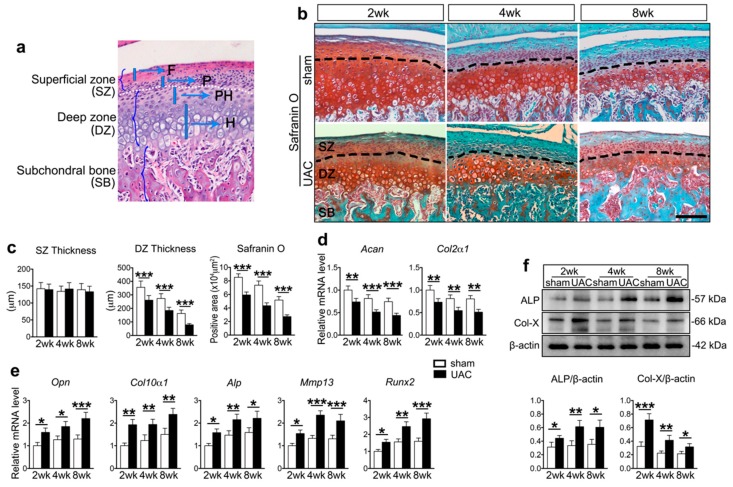
The changes in histomorphology of the unilateral anterior crossbite (UAC) rat temporomandibular joint (TMJ) cartilage. (**a**) Sagittal central section of the TMJ condylar cartilage showing the four cellular layers. The fibrous (*F*) and proliferative layers (*P*) make up the superficial zone. The prehypertrophic (*PH*) and hypertrophic layers (*H*) make up the deep zone. *F*, fibrous layer. *P*, proliferative layer. *PH*, prehypertrophic layer. *H*, hypertrophic layer. (**b**) Representative sagittal central sections with Safranin O staining of the UAC and age matched sham control TMJs at 2, 4, and 8 weeks. Magnification, 200×. Scale bar, 100 μm. Black dotted lines distinguish the superficial and deep zone of the cartilage. *SZ*, superficial zone. *DZ*, deep zone. *SB*, subchondral bone. (**c**) Comparison of the thickness of the superficial zone and deep zone cartilage and the Safranin O positive area between UAC and the age-matched sham control groups. *n* = 6. (**d**,**e**) The mRNA expression levels of genes relate to chondrogenic matrix (**d**) and terminal differentiation (**e**) in the TMJ cartilage obtained from rats exposed to 2, 4, and 8 weeks UAC treatment and their age matched sham controls. *n* = 3. (**f**) Western blot of TMJ cartilage for ALP and Col-X, and the related quantification. Results are represented as the mean ± SD. * *p* < 0.05, ** *p* < 0.01, *** *p* < 0.001.

**Figure 2 ijms-20-03797-f002:**
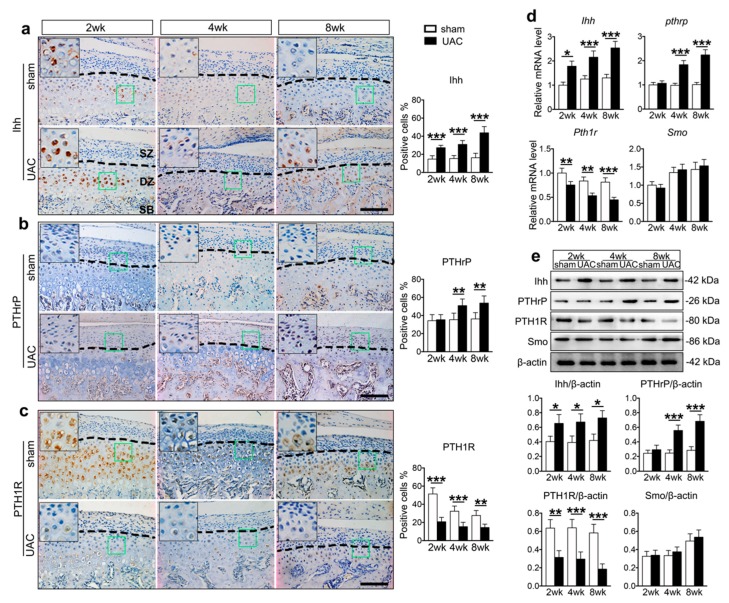
The changes of Indian hedgehog (Ihh) and parathyroid hormone receptor 1 (PTHrP) expression levels in TMJ cartilage of the UAC treated rat. (**a**–**c**) Representative sagittal central sections and the quantification of the positive cells with immunohistochemical staining in UAC and sham TMJ cartilage from 2, 4, and 8 weeks. The green box regions in the images were magnified at the top left corner. Black dotted lines distinguish the superficial and deep zone chondrocyte. *SZ*, superficial zone. *DZ*, deep zone. *SB*, subchondral bone. Magnification, 200×. Scale bar, 100 μm. *n* = 6. (**d**) The mRNA expression levels of *Ihh*, *Pthrp*, *Smo*, and *Pth1r* in the UAC and sham TMJ cartilage at 2, 4, and 8 weeks. *n* = 3. (**e**) Western blot of pure cartilage tissues harvested from TMJ for Ihh, PTHrP, Smo, and PTH1R and the related quantification. *n* = 3. Results are represented as the mean ± SD. * *p* < 0.05, ** *p* < 0.01, *** *p* < 0.001.

**Figure 3 ijms-20-03797-f003:**
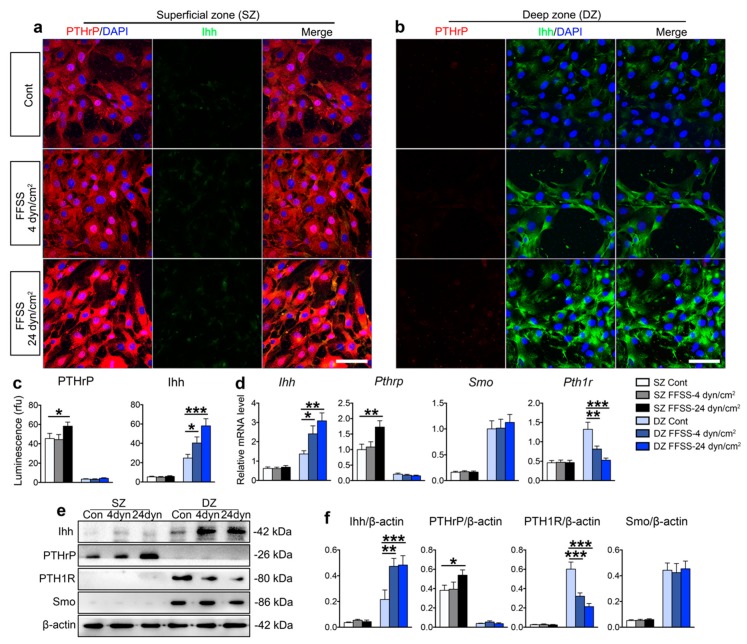
Ihh and PTHrP signaling changes in the superficial and deep zone cells under FFSS. (**a**,**b**) Immunofluorescence staining of PTHrP and Ihh in the cultured superficial (**a**) and deep zone (**b**) cells obtained from the mandibular condylar cartilage of the 3-week-old rats. FFSS was applied at 4 and 24 dyn/cm^2^. Cont: Cells were not treated by FFSS and taken as controls (**c**) The quantification of immunofluorescence intensity for PTHrP and Ihh. (**d**–**f**) Comparison of mRNA (**d**) and protein expression (**e**,**f**) of Ihh, PTHrP, Smo, and PTH1R. Magnification, 400×. Scale bar, 25 μm. *n* = 3. Results are represented as the mean ± SD. * *p* < 0.05, ** *p* < 0.01, *** *p* < 0.001.

**Figure 4 ijms-20-03797-f004:**
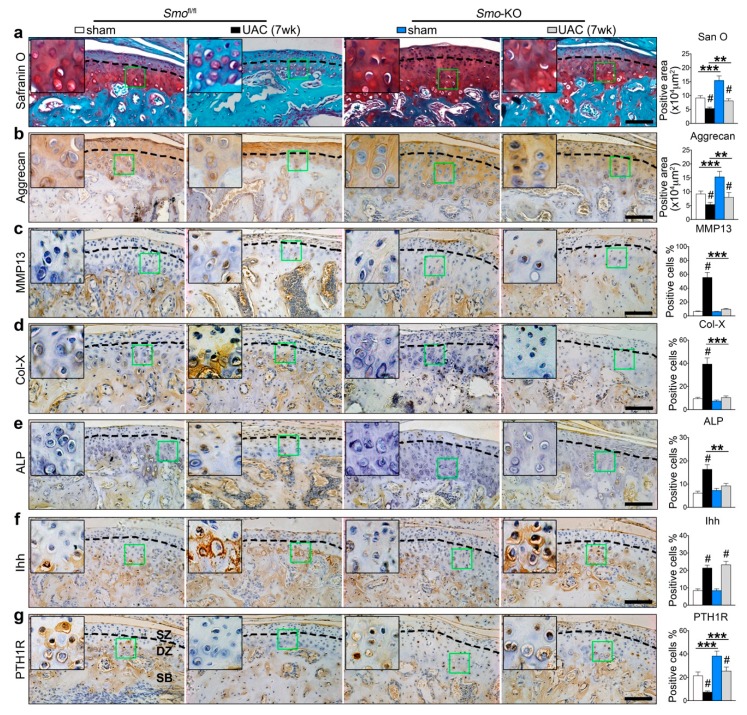
Reduced cartilage matrix and enhanced terminal differentiation in UAC group were reversed by inhibiting Ihh signaling through deleting *Smo*. Cartilage-specific and tamoxifen-inducible *Smo* gene knockout (*Smo*-KO) mice were used. The *Smo*^fl/fl^ mice were used as the genetic control. UAC was applied to 6-week-old mice for 7 weeks. Safranin O staining (**a**), immunohistochemical staining of aggrecan (**b**), MMP-13 (**c**), Col-X (**d**), ALP (**e**), Ihh (**f**), and PTH1R (**g**) and the quantitative data (right panels) are presented. The green box regions in the images were magnified at the top left corner. Black dotted lines distinguish the superficial and deep zone chondrocyte. *SZ*, superficial zone. *DZ*, deep zone. *SB*, subchondral bone. Magnification, 200×. Scale bar, 100 μm. *n* = 6. Results are represented as the mean ± SD. ** *p* < 0.01, *** *p* < 0.001 for the differences between *Smo*-KO group and *Smo*^fl/fl^ control group. # *p* < 0.05 for the differences between the sham and UAC group.

**Figure 5 ijms-20-03797-f005:**
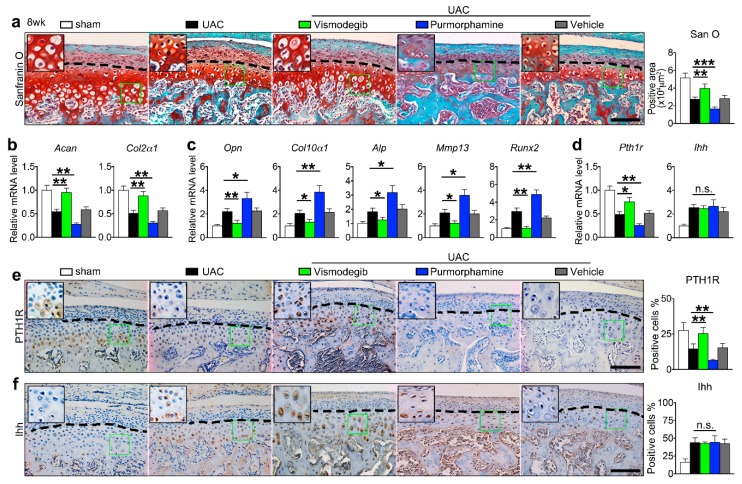
The effect of drug-modified Ihh signaling on UAC stimulated deep zone chondrocytes terminal differentiation in TMJ cartilage of rats. Vismodegib or purmorphamine (1 μM) was injected in TMJ region from 4 to 8 weeks after UAC treatment. (**a**) Safranin O staining of TMJ and the quantitative data (right panel). (**b**,**c**) Comparison of mRNA expression of cartilage matrix and terminal differentiation markers. (**d**) The mRNA expression of *Pth1r* and *Ihh*. (**e**) Immunohistochemical staining for PTH1R and its quantification of percentage of the positive cells (right panel). (**f**) Immunohistochemical staining of Ihh and its quantification of percentage of the positive cells (right panel). The green box regions in the images were magnified at the top left corner. Black dotted lines distinguish the superficial and deep zone chondrocyte. *SZ*, superficial zone. *DZ*, deep zone. *SB*, subchondral bone. Magnification, 200×. Scale bar, 100 μm. IHC staining: *n* = 6. qPCR analysis: *n* = 3. Results are represented as the mean ± SD. * *p* < 0.05, ** *p* < 0.01, *** *p* < 0.001.

**Figure 6 ijms-20-03797-f006:**
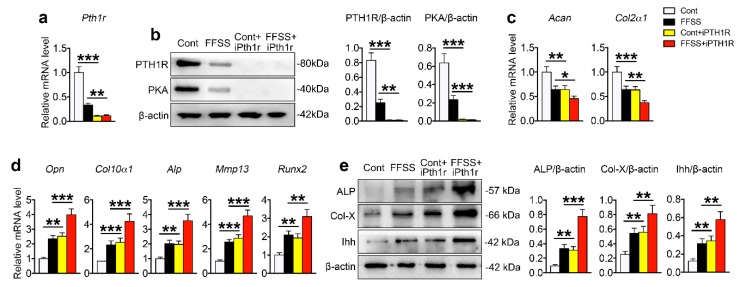
PTH1R expression level in the deep zone cells of TMJ cartilage influenced the terminal differentiation. The deep zone cells were isolated from the mandibular condylar cartilage of the 3-week-old SD rats and were transfected with *Pth1r* siRNA to knockdown its expression. (**a**) The expression of *Pth1r*. (**b**) The protein expression of PTH1R and PKA. (**c**,**d**) The expression of *Acan*, *Col2a1*, *Opn*, *Col10a1*, *Alp*, *Mmp13*, and *Runx2*. (**e**) The protein expression of ALP, Col-X, and Ihh. *n* = 3. Results are represented as the mean ± SD. * *p* < 0.05, ** *p* < 0.01, *** *p* < 0.001.

**Figure 7 ijms-20-03797-f007:**
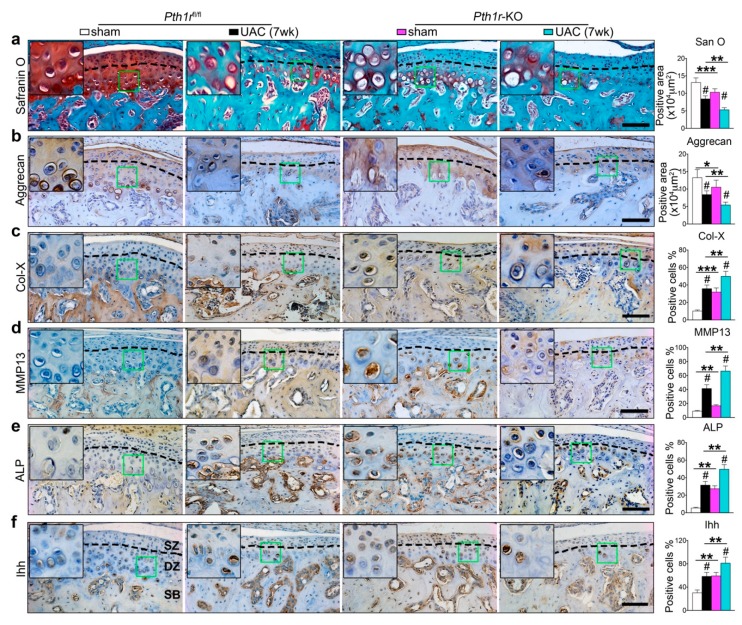
*Pth1r* knockout promoted the deep zone chondrocyte terminal differentiation. Cartilage-specific and tamoxifen-inducible *Pth1r* gene knockout (*Pth1r*-KO) mice were exposed to 7 weeks of UAC treatment. The *Pth1r*^fl/fl^ mice were used as the genetic control. (**a**) Safranin O staining and the quantitative data (right panel). (**b**–**f**) Immunohistochemical staining and the quantitative data (right panel) for aggrecan (**b**), Col-X (**c**), MMP-13 (**d**), ALP (**e**), and Ihh (**f**). The green box regions in the images were magnified at the top left corner. Black dotted lines distinguish the superficial and deep zone chondrocyte. *SZ*, superficial zone. *DZ*, deep zone. *SB*, subchondral bone. Magnification, 200×. Scale bar, 100 μm. *n* = 6. Results are represented as the mean ± SD. * *p* < 0.05, ** *p* < 0.01, *** *p* < 0.001 represent significant differences between knockout group and genetic control group. # *p* < 0.05 represent significant differences between the sham and UAC group in the *Pth1r*^fl/fl^ and knockout groups.

**Figure 8 ijms-20-03797-f008:**
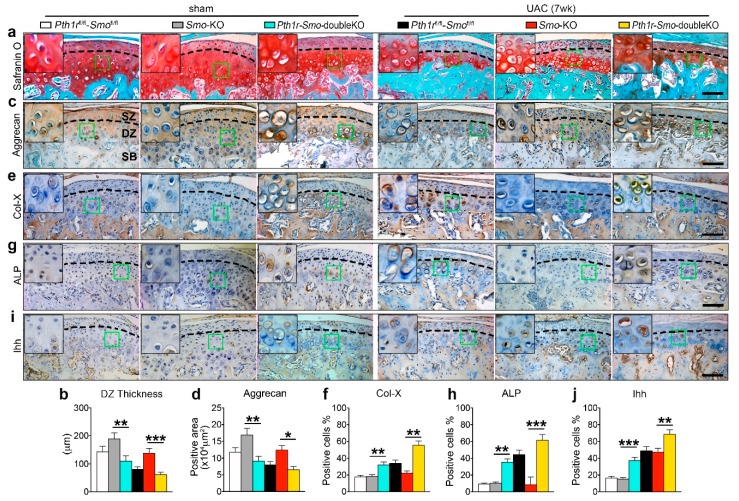
PTH1R dependent effect of Ihh signaling on UAC induced TMJ chondrocytes terminal differentiation. Cartilage-specific and tamoxifen-inducible *Pth1r* and *Smo* gene double knockout mice (*Pth1r-Smo*-double KO) were treated with UAC for 7 weeks. The *Smo*^fl/fl^-*Pth1r*^fl/fl^ mice were used as a blank control and *Smo* knockout as a positive control. (**a**,**b**) Safranin O staining and its quantification. (**c**–**j**) Immunohistochemical staining and the quantification of aggrecan (**c**,**d**), Col-X (**e**,**f**), ALP (**g**,**h**), and Ihh (**i**,**j**). The green box regions in the images were magnified at the top left corner. Black dotted lines distinguish the superficial and deep zone chondrocyte. *SZ*, superficial zone. *DZ*, deep zone. *SB*, subchondral bone. Magnification, 200×. Scale bar, 100 μm. *n* = 6. Results are represented as the mean ± SD. * *p* < 0.05, ** *p* < 0.01, *** *p* < 0.001.

**Figure 9 ijms-20-03797-f009:**
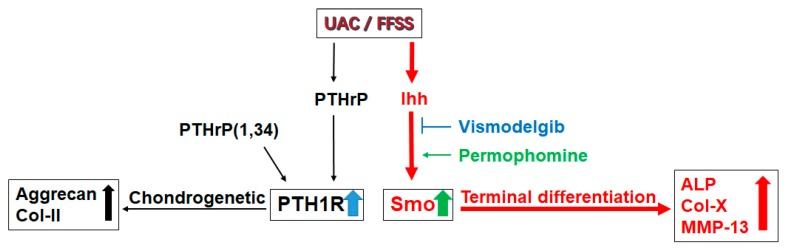
The summary of the present story. Both UAC and flow fluid shear stress (FFSS) stimulated section of PTHrP and Ihh, and promoted chondrocytes differentiation, which reduced the number of PTH1R expressing chondrocytes.

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
