# Peer review of "Inhibition of Ihh Reverses Temporomandibular Joint Osteoarthritis via a PTH1R Signaling Dependent Mechanism"

_ijms, 2019, doi:10.3390/ijms20153797_

Round 1

Reviewer 1 Report

The article has been written in a beautiful way and the data supports the conclusion of the study. Few grammatical errors which can be improved. Secondly, in the histological pictures authors should mention the magnification of the images so that it becomes easy for the reareds to read the article.

Author Response

1. The article has been written in a beautiful way and the data supports the conclusion of the study. Few grammatical errors which can be improved.

Response 1: Thank you for your comments. We have improved our grammatical errors in the revised manuscript by the English Editing serves from MDPI.

2. Secondly, in the histological pictures authors should mention the magnification of the images so that it becomes easy for the readers to read the article.

Response 2: We have added the magnification of every figures for readable.

Reviewer 2 Report

Although this is a very interesting manuscript but has some scientific and technical challenges

For example- "Vismodegib (1 M) or purmorphamine (1 M) was injected every other day from 4 to 8 weeks after 355 UAC for a total of 14 times within 4 weeks". 

The continuous injection14 times could induce severe inflammation and promote OA- related inflammatory cytokines specifically IL-1b, TNF-alpha and others. It is difficult to evaluate the effect of this type of local inflammatory environment.

The manuscript is very difficult to understand as the per presented data.

Most of the conclusions are not supported by the data presented. The final figure with an overview is not supported completely by the presented data.

Since it has enough experiment to justify signaling network, it should be presented in the logical progression. Some data could be provided as a supplementary instead of being main text.

Author Response

1. Although this is a very interesting manuscript but has some scientific and technical challenges. For example "Vismodegib (1 mM) or purmorphamine (1 mM) was injected every other day from 4 to 8 weeks after UAC for a total of 14 times within 4 weeks". The continuous injection 14 times could induce severe inflammation and promote OA-related inflammatory cytokines specifically IL-1b, TNF-alpha and others. It is difficult to evaluate the effect of this type of local inflammatory environment.

Response1: Articular cavity injection of drugs or inhibitors are adopted in clinical treatment for OA disease. Here, we injected the Ihh inhibitor into the local area of rat TMJ to cure UAC induced OA. We used the insulin syringe to execute the locally injection. Previously, we had injected BMSCs (1) and TNF inhibitors (2) using the same methods, and so long we had not observed significant local inflammatory reaction caused simply by injection procedure. We had detected the expression of TNF-alpha and IL-1beta, and did not noticed any significant changes in the expression levels [ref #23]. We have added more details in the Results and Discussion section. (line 216-217, 405-410)

Ref:

1. Zhang, M.; Yang, H.; Lu, L.; Wan, X.; Zhang, J.; Zhang, H.; Liu, X.; Huang, X.; Xiao, G.; Wang, M. Matrix replenishing by BMSCs is beneficial for osteoarthritic temporomandibular joint cartilage. Osteoarthritis. Cartilage. 2017, 25, 1551-1562.

2. Yang, H.X.; Zhang, M.; Wang, X.; Zhang, H.; Zhang, J.; Jing, L.; Liao, L.; Wang, M. TNF Accelerates Death of Mandibular Condyle Chondrocytes in Rats with Biomechanical Stimulation-Induced Temporomandibular Joint Disease. PLoS. One. 2015, 10, e0141774.

2. The manuscript is very difficult to understand as the per presented data. Most of the conclusions are not supported by the data presented. The final figure with an overview is not supported completely by the presented data.

Response2: We have revised our manuscript to make the conclusions in results section more readable and easy understand. Also, the final figure has been replaced by a simple overview for our presented data.

3. Since it has enough experiment to justify signaling network, it should be presented in the logical progression. Some data could be provided as a supplementary instead of being main text.

Response3: We have exchanged the results part 3 and part 4. For more logical progression, we firstly described the UAC and FFSS stimulated terminal differentiation. We then expressed that suppression of Ihh signaling reversed the UAC induced chondrocyte terminal differentiation through a PTH1R dependent mechanism. We believe that the revised manuscript could be more logical for read.

Round 2

Reviewer 2 Report

Although authors made changes to the earlier version, it should be simplified further and presented as easy to understand.

There are some typos which should be corrected.

Author Response

1. Although authors made changes to the earlier version, it should be simplified further and presented as easy to understand.

Response1: We have simplified our manuscript for an easy read.

In discussion section, the original expression that “This assumption was confirmed in double knockout mice. Additional deletion of Pth1r abrogated the rescue effect of Smo deletion on UAC-stimulated cartilage degradation. Obviously, the rescue effect of Ihh inhibition on OA lesions is PTH1R dependent.” Have been simplified as “This assumption was confirmed by the fact that additional deletion of Pth1r abrogated the rescue effect of Smo deletion on UAC-stimulated cartilage degradation.”

We also simplified the Figure legends, especially in the supplemental Figure legends.

2. There are some typos which should be corrected.

Response2: Thank you for your comments. We have improved typos in the revised manuscript by the English Editing serves from MDPI.